# Cloning, Identification, and Functional Analysis of the *Chalcone Isomerase* Gene from *Astragalus sinicus*

**DOI:** 10.3390/genes14071400

**Published:** 2023-07-05

**Authors:** Xian Zhang, Jing Xu, Linlin Si, Kai Cao, Yuge Wang, Hua Li, Jianhong Wang

**Affiliations:** 1Institute of Environment, Resource, Soil and Fertilizer, Zhejiang Academy of Agricultural Sciences, Hangzhou 310021, China; 2College of Science, Northeastern University, Boston, MA 02115, USA; wang.yug@northeastern.edu

**Keywords:** *Astragalus sinicus*, *chalcone isomerase* (*CHI*), anthocyanin, gene expression, characterization

## Abstract

*Astragalus sinicus* is an important winter-growing cover crop. It is widely utilized, not only as a cover crop for its benefits in fertilizing the soil but also as a landscape ground cover plant. Anthocyanins are involved in the pigmentation of plants in leaves and flowers, which is a crucial characteristic trait for *A. sinicus.* The formation of anthocyanins depends significantly on the enzyme chalcone isomerase (CHI). However, research on the *CHI* gene of *A. sinicus* remains unexplored. The rapid amplification of cDNA ends (RACE) approach was used in this research to clone the *CHI* sequence from *A. sinicus (AsiCHI*). The expression profiles of the *AsiCHI* gene in multiple tissues of *A. sinicus* were subsequently examined by qRT-PCR (Quantitative Real-Time PCR). Furthermore, the function of the *AsiCHI* was identified by the performance of ectopic expression in *Arabidopsis* (*Arabidopsis thaliana*). The outcomes revealed that the full-length cDNA of the *AsiCHI* gene (GeneBank: OQ870547) measured 972 bp in length and included an open reading frame of 660 bp. The encoded protein contains 219 amino acids with a molecular weight of 24.14 kDa and a theoretical isoelectric point of 5.11. In addition, the remarkable similarity between the AsiCHI protein and the CHI proteins of other *Astragalus* species was demonstrated by the sequence alignment and phylogenetic analysis. Moreover, the highest expression level of *AsiCHI* was observed in leaves and showed a positive correlation with anthocyanin content. The functional analysis further revealed that the overexpression of *AsiCHI* enhanced the anthocyanidin accumulation in the transgenic lines. This study provided a better understanding of *AsiCHI* and elucidated its role in anthocyanin production.

## 1. Introduction

Anthocyanins, a category of water-soluble pigment, are generated through a branch of the flavonoid pathway and are responsible for the majority of the red, blue, and purple colors seen in plant leaves, fruits, and flowers [1,2]. Besides their role in promoting visual aesthetic appeal in natural and managed landscapes, anthocyanins also protect plants from a range of stresses, such as UV radiation, temperature fluctuations, insects, herbivores, and fungal infections [3,4]. In addition, anthocyanins show many medicinal values regarding human health benefits, especially in preventing cardiovascular diseases, promoting night vision, and alleviating oncogenicity [5,6]. Anthocyanins’ complex features underline their importance in both plant physiology and human health, making them a subject of substantial scientific interest and research.

Chalcone isomerase (CHI) is an enzyme that plays an important role in the synthesis of flavonoids, including proanthocyanidins and anthocyanins [7]. CHI, the second rate-limiting enzyme in the flavonoid pathway, is involved in a range of plant growth, stress, and defense responses. It catalyzes the intramolecular cyclization process, transforming the bicyclic chalcone into tricyclic (2S)-flavanone, which is the exclusive substrate for downstream enzymes [8]. Previous research has shown that mutations with a deletion of the *CHI* gene can result in a considerable drop in the number of anthocyanins generated, and the expression level of *CHI* genes can significantly impact anthocyanin accumulation. [9]. For example, in *Petunia hybrida*, the low *CHI* expression levels in white petals reduced pelargonidin (PelC) production. In contrast, high expression levels of *CHI* and anthocyanin production were observed in red petals [10]. Furthermore, the *CHI* gene in *G. biloba L*. was transformed into *E. coli*, and the activity of *CHI* showed a strong correlation with the flavonoid content [11]. Additionally, novel *CmCHI* genes from *Clivia miniata* were discovered, which were shown to be strongly expressed in anthocyanin-accumulating tissues [12]. Moreover, *CHI* co-expression may also have a profound influence on the deposition of flavonoids [13]. Co-expression of two transcription factors, the *Delilah* (*Del*)/*Rosea1* (*Ros1*) gene and the *CHI* gene, in tomato plants resulted in considerably higher anthocyanin content compared to all transgenic tomatoes containing only *CHI* or *Del/Ros1* [14]. Thus, genetic manipulation of *CHI* expression is a promising approach for modulating flavonoid/anthocyanin content in plants.

Chinese milk vetch, also known as *A. sinicus*, is a winter cover crop that has been extensively employed in regions with temperate climates, including southern China, Korea, and Japan [15]. In addition, *A. sinicus* serves as a landscape ground cover plant for the sea of flowers, fodder for animals, a source of honey for bees, and an important Chinese traditional herb [16]. As a leguminous cover crop, *A. sinicus* plays a significant role in biological nitrogen fixation, thereby contributing to nitrogen input in cropping systems [17,18]. Rotation of *A. sinicus* during fallow periods can effectively exploit natural resources such as light, water, and heat, leading to sustainable improvements in agroecosystems [15]. However, the scarcity of phosphorus (P) frequently limits the ability to fix nitrogen [19,20]. Anthocyanin content is an indicator in plants, as *A. sinicus* cultivars with higher anthocyanin content are more tolerant to low-P stress [21,22]. Moreover, anthocyanins are involved in the pigmentation of leaves and flowers [22], which is an important trait for *A. sinicus* as a landscape ground cover plant. Hence, it is essential to reveal the pathway involved in the production of anthocyanin in *A. sinicus.*

This article presents the initial report on the cloning and characterization of a novel chalcone isomerase gene obtained from *A. sinicus*. The cDNA of *chalcone isomerase* in *A. sinicus* was successfully isolated. Indeed, a comprehensive analysis was carried out to examine the expression pattern of the *AsiCHI* gene across various tissues of *A. sinicus*. In addition, the role of the *AsiCHI* gene in anthocyanin biosynthesis was identified through the overexpressed lines of *Arabidopsis.* The findings suggested that *AsiCHI* may be crucial in boosting the *A. sinicus* plant’s ability to produce anthocyanins. It provides a scientific basis for further studies on the molecular biological function of the anthocyanin biosynthesis pathway in *A. sinicus*.

## 2. Materials and Methods

### 2.1. Plant Materials

The selected white petal *A. sinicus* population was meticulously cultivated in the greenhouse of the Zhe Jiang Academy of Agricultural Sciences, Hang Zhou, Zhe Jiang, China. The transgenic *Arabidopsis* plants were maintained in a controlled environment (25 °C, 16 h light/8 h dark) until harvesting. Fresh and clean *Arabidopsis* seedlings and plant tissues of *A. sinicus*, such as stems, flowers, flower buds, leaves, and roots, were sampled with three biological repeats, respectively, for evaluating relative gene expression and anthocyanin content. Upon collection, the samples were promptly submerged in liquid nitrogen and subsequently stored at a temperature of −80 °C for future utilization.

### 2.2. Extraction of Total RNA and Genomic DNA

Genomic DNA and total RNA were extracted from plant tissues with the Super Plant Genomic DNA Kit (Tiangen Biotech, Beijing, China) and the FastPure Universal Plant Total RNA Isolation Kit (Vazyme, Nanjing, China), following the manufacturer’s instructions. Spectrophotometer analysis and agarose gel electrophoresis were then implemented to evaluate the quality and concentration of the isolated DNA and RNA.

### 2.3. Cloning of CHI and Bioinformatics Analyses

Transcriptome analysis was conducted to obtain genes in response to anthocyanin biosynthesis. One of the cDNA fragments was predicted to encode *chalcone isomerase*. Based on this sequence, gene-specific primers were created to amplify the full-length cDNA sequence of *CHI* from *A. sinicus*. 5′ and 3′ RACE PCR methods were applied to acquire the *A. sinicus CHI*. The first-strand cDNAs were synthesized from *A. sinicus* RNA and used as templates for both 5′ and 3′ end sequence cloning. According to the manufacturer’s instructions, the 3′-cDNA and 5′-cDNA were performed by GeneRacer Kit (Invitrogen, CA, USA). Then the full-length cDNA sequence was obtained by assembling the original cDNA fragment with its 5′ and 3′ cDNA ends. Primer data are listed in Table 1.

The NCBI website (www.ncbi.nlm.nih.gov, accessed on 6 April 2023) was leveraged to evaluate the nucleotide sequences, deduced amino acid sequences, open reading frame (ORF), and homologous sequences. The ProtParam software (http://web.expasy.org/protparam/, accessed on 6 April 2023) was employed to predict the physicochemical properties of CHI, which encompassed the determination of its amino acid sequence, relative molecular weight, and isoelectric point. The multiple alignments of CHI were aligned with ClustalX, and the phylogenetic tree was generated with 1000 bootstraps by MEGA 6.0. The protein sequence of *A. sinicus* L. *Chalcone isomerase* was compared with orthologous sequences from other representative species. The Genbank accession numbers are as follows: *A. sinicus* L. (OQ870547), *Astragalus membranaceus* (ATY39974.1), *Astragalus mongholicus* (ABA55017.2), *Glycine max* (Q53B70.1), *Pueraria montana* var. *lobata* (ADV71377.1), *Pisum sativum* (XP_050892596.1), *Vigna unguiculata* (QCE05742.1), and *Medicago sativa* (AAB41524.1).

### 2.4. Quantitative Real-Time PCR (qRT-PCR) and Determination of Anthocyanins Content

By using the FQD-48A(A4) real-time PCR detection system (BIOER, Hangzhou, China) and Power SYBR^®^ Green PCR Master Mix, quantitative real-time PCR was performed to measure the degree of *CHI* gene expression across several *A. sinicus* tissues. The first-strand cDNAs were synthesized from *A. sinicus* RNA using Hifair^®^ II 1st Strand cDNA Synthesis SuperMix for qPCR (Yeasen, Shanghai, China). Two specific primers were designed according to the sequence of *AsiCHI* (Table 1). The reference gene employed in this study was *β-tubulin* from *A. sinicus* (GeneBank: OR198065) Each qPCR reaction was conducted in a 20 μL total volume, comprising specific components including SYBR Premix Ex Taq, forward and reverse primers, cDNA template, and ddH_2_O. The thermal cycling conditions consisted of an initial step at 95 °C for 30 s, followed by 40 cycles of specific durations at 95 °C and 60 °C for 20 s. Three replicates of each reaction were carried out, and the 2^−ΔΔCt^ method was applied for calculating the relative fold change of gene expression in qRT-PCR. 

Six kinds of anthocyanins including Cyanidin (CC), Delphinidin (DC), Pelargonidin (PelC), Peonidin (PeoC), Petunidin (PT), and Malvidin (MV) in different *A. sinicus* tissues were identified and quantified by HPLC-MS/MS analysis, using the modified Wu et al. method [5]. 

### 2.5. Transformation and Phenotypic Analysis of Transgenic Arabidopsis

The transformation of *Arabidopsis* ecotype Columbia (Col-0) was performed using the floral dip method. After being inserted into the pCAMBIA1302 vector, the full-length *AsiCHI* cDNA was sequenced to ensure its accuracy. Seeds of different transgenic lines were placed on ½-strength MS medium, which contains 50 mg/L hygromycin. Then the hygromycin-resistant transgenic plants were tested by PCR analysis. Homozygous transgenic progeny lines were obtained for further experiments through the kanamycin-resistance test and PCR analysis. 

The interrelationship between *AsiCHI* expression level and anthocyanin accumulation in the transgenic system was detected. Seedings of wild-type and transgenic *Arabidopsis* were harvested for qRT-PCR and anthocyanin content analysis, following the protocols described above. 

### 2.6. Statistical Analysis

All of the above data were repeated in three biological replicates and analyzed using SPSS 26.0 (SPSS, Chicago, IL, USA). 

## 3. Results

### 3.1. Full-Length Cloning of AsiCHI and Sequence Analysis

A *CHI* gene was first obtained from *A. sinicus* by the rapid amplification of cDNA ends (RACE) method and submitted to NCBI (accession number: OQ870547). The *AsiCHI* gene had a full length of 972 bp, with the 5′ and 3′ untranslated regions (UTR) of 131 bp and 181 bp, respectively. The gene contains an open reading frame (ORF) of 660 nucleotide sequences, which encodes a protein of 219 amino acids. The predicted molecular weight of the *AsiCHI* protein is 24.14 kDa with an isoelectric point (pI) of 5.11 (Table 2).

### 3.2. Sequence Alignment and Phylogenetic Analysis of AsiCHI Protein

Sequence alignment analysis of the AsiCHI protein and ten sequences from well-characterized orthologous proteins from various species (*Astragalus membranaceus, Astragalus mongholicus*, *Glycine max*, *Pueraria montana* var. *lobata, Pisum sativum*, *Vigna unguiculata*, *Medicago sativa*) was performed using Clustal X 1.83. The several alignments demonstrated that *A. sinicus* and the examined sequences were exceptionally comparable, ranging from 59.72% to 82.65% (Table 2). The AsiCHI protein presented the highest similarity with *Astragalus membranaceus* (82.65%) and *Astragalus mongholicus* (82.19%) while exhibiting the lowest similarity with *Medicago sativa* (59.72%) (Table 2). 

To further analyze the relationship between various proteins, protein sequences were aligned using the Mega X program to create a phylogenetic tree, and a bootstrap test with 1000 iterations was used to determine the tree’s credibility. The result revealed that *A. sinicus, Astragalus membranaceus*, and *Astragalus mongholicus* clustered closely together in the same branch due to their close evolutionary relationship (Figure 1). These genes’ placement in the phylogenetic tree was compatible with where they had evolved.

Additionally, the AsiCHI protein contained 13 active amino acid sites: Arg^36^, Gly^37^, Leu^38^, Phe^47^, Thr^48^, Ile^50^, Tyr^106^, Lys^110^, Val^111^, Asn^113^, Cys^114^, Thr^190^, and Met^191^. These amino acid residues are found to be highly conserved in all the CHI proteins analyzed in this study (Figure 2).

### 3.3. Gene Expression Profiles of AsiCHI by qRT-PCR

The *AsiCHI* transcriptional expression in different tissues (roots, stems, leaves, flower buds, and flowers) of *A. sinicus* was determined by qRT-PCR. The outcome proved that *AsiCHI* expression was identified in all tissues examined, with leaves expressing it at the highest level and flower buds at the lowest level (Figure 3). An extremely high expression level of the *AsiCHI* gene was observed in leaves, but no significant difference was found among the expression levels of roots, stems, flower buds, and flowers of *A. sinicus*.

### 3.4. The Content of Anthocyanidins

Anthocyanidins were isolated from a variety of *A. sinicus* tissues, including the roots, stems, leaves, flower buds, and flowers, in order to investigate the association between gene expression and anthocyanin synthesis in this plant. HPLC-MS/MS was used to detect the content of six anthocyanidins, including cyanidin (CC), delphinidin (DC), pelargonidin (PelC), peonidin (PeoC), petunidin (PT), and malvidin (MV). The results demonstrated that these six anthocyanidins were all detected in the tested tissues of *A. sinicus*. The examined anthocyanidins showed significantly higher accumulation in leaves compared to roots, stems, flower buds, and flower tissues. The total anthocyanidin content in leaves (689.75 μg/g) was about two to three times higher than in roots (211.97 μg/g), stems (268.69 μg/g), flower buds (241.68 μg/g), and flowers (263.29 μg/g) (Figure 4). The levels of *AsiCHI* expression were significantly positively correlated with anthocyanin concentration (Table 3).

### 3.5. Anthocyanidins Analysis in Transgenic Arabidopsis

To examine the function of the *AsiCHI* gene in *A. sinicus*, the cDNA was overexpressed in *Arabidopsis* Col-0 (wild-type WT). Here, 55 independent homozygous transgenic lines were obtained through the kanamycin-resistance test and PCR analysis. The *AsiCHI* gene was expressed in transgenic lines but not detected in wild-type seedlings. In order to delve deeper into the impact of *AsiCHI* overexpression on anthocyanin production, the levels of anthocyanidins were quantitatively analyzed in both *AsiCHI-*overexpression transgenic lines and the wild-type using HPLC-MS/MS techniques. In accordance with the *AsiCHI* gene expression levels, the results illustrated that the contents of six anthocyanins in the *AsiCHI*-overexpression seedlings were significantly higher than those in the wild-type (Figure 5). 

## 4. Discussion

Anthocyanin is a flavonoid substance that serves a vital role in various biological functions, including color formation and stress response [23,24]. In addition to giving *A. sinicus*’ leaves and flowers their distinctive purple color, anthocyanin also protects plants from various stresses, such as low P-stress [21,22]. Due to its crucial biological functions, increasing the amount of anthocyanin and/or flavonoids in *A. sinicus* has been a significant research objective. However, anthocyanin metabolism was poorly understood in *A. sinicus*. The involvement of chalcone isomerase (CHI) in this context is of particular relevance since it is an indispensable enzyme in the generation of anthocyanins/flavonoids and the second rate-limiting enzyme in their biosynthetic pathway [25]. *A. sinicus* was used in this investigation to clone a novel *CHI* gene called *AsiCHI*. The *AsiCHI* gene was identified and characterized through sequence alignment, tissue expression distribution, phylogenetic tree construction, and function analysis.

According to their phylogenetic relationships and functions, CHIs can be classified into four types [26]. Type I CHIs are commonly found in vascular plants and possess the ability to specifically catalyze the transformation of naringenin chalcone into naringenin [27]. Type II CHIs, which are more common in ancient land plants, such as Selaginella and liverworts than Type I CHIs, which are primarily observed in leguminous plants [28]. They can catalyze both isoliquiritigenin and naringenin chalcone, resulting in the formation of liquiritigenin and naringenin, respectively [29]. Type III CHIs, also known as fatty-acid-binding proteins (FAPs), are associated with fatty acid metabolism [13]. Furthermore, chalcone isomerase-like (CHIL) proteins, commonly referred to as Type IV CHIs, lack some of CHI’s catalytic properties and are involved in plant flavonoid biosynthesis as enhancers or rectifiers [30,31,32]. Previous studies have discovered that the deduced amino acid sequences of CHIs of the same type from different species showed a high identity of about 70%, while the similarity between different types was only about 50% [26]. The sequence alignment analysis in this research illustrated a significant alignment between the deduced polypeptide sequence of AsiCHI and the well-established type II CHI of Glycine max, demonstrating an overall identity of 70.64% to Glycine max CHI (Table 1) [33,34]. In addition, the combination of two amino acid residues at positions 190 and 191 (the residue numbering is based on the *MsCHI* sequence scheme) was postulated to determine substrate preference and found to be clearly different between type I and type II CHIs [27,35]. Residues at positions 190 and 191 of type II CHI were Thr and Met [8]. In this paper, the AsiCHI ortholog sequences contain 13 active amino acid sites: Arg^36^, Gly^37^, Leu^38^, Phe^47^, Thr^48^, Ile^50^, Tyr^106^, Lys^110^, Val^111^, Asn^113^, Cys^114^, Thr^190^, and Met^191^. When compared to known type II CHIs, they have several similar residues, such as Thr^190^ and Met^191^ [8] (Figure 2). Therefore, the results illustrated that the *AsiCHI* gene encodes a type II *CHI* and is essential in the anthocyanin pathway.

Previous investigations on the diversity of *CHI* gene expression in different plant tissues and their distinctive physiological roles have shown that gene expression patterns are strongly associated with their particular functions [36,37]. To elucidate the physiological role of *AsiCHI* in anthocyanin biosynthesis, the expression profile of *AsiCHI* was evaluated by qRT-PCR in different *A. sinicus* tissues. Notably, previous research has shown that leguminous type II *CHI*s exhibit root-specific expression and can act as signaling molecules for root-nodule development [38]. In contrast, the *AsiCHI* gene was ubiquitously expressed in tested tissues, with an extremely high expression level observed in leaves. The significantly low expression of *AsiCHI* in roots indicated that *AsiCHI* probably does not respond to bradyrhizobium inoculation but plays a critical role in the formation of anthocyanin.

Six different anthocyanidins, including cyanidin (CC), delphinidin (DC), pelargonidin (PelC), peonidin (PeoC), petunidin (PT), and malvidin (MV), were additionally found in various *A. sinicus* tissues in this experiment to further support the correlation between gene expression and anthocyanin synthesis. All the examined anthocyanidins accumulated significantly higher in leaves compared to roots, stems, flower buds, and flowers. This demonstrated a strong consistency between the expression of *AsiCHI* transcripts in different tissues and the accumulation of anthocyanins, further verifying its function in anthocyanin biosynthesis.

Previous studies have highlighted the significant influence of *CHI* gene expression on flavonoid and anthocyanin biosynthesis in plants. Overexpression of the *CHI* gene in transgenic plants has been shown to effectively enhance the content of flavonoids [9]. For instance, through the cloning of a *FtCHI* gene from *Tartary buckwheat* and subsequent overexpression in transgenic *Arabidopsis*, it was observed that the total flavonoid content in the transgenic *Arabidopsis* plants was notably higher compared to the wild-type counterparts [39]. Similarly, flavonoid accumulation was enhanced by overexpressing *CtCHI* genes from *Carthamus tinctorius* in transgenic *Arabidopsis* [9]. By introducing sweet potato *IbCHI* into the *Arabidopsis tt5* mutant, researchers were able to restore the seed coats’ pigmentation as well as the purple hue of the cotyledons and hypocotyls, demonstrating the importance of *CHI* in the formation of flavonoids and anthocyanins [40]. The transfer of the *Solanum melongena SmCHI* gene into *Arabidopsis* resulted in increased anthocyanin content and visible pigments in the stems and siliques of *Arabidopsis* transgenic plants [41]. The overexpression of the *CHI* gene from the onion *Allium cepa* in tomatoes displayed an increase in flavonol and anthocyanin content in the peel [14]. By inserting the overexpression *Paeonia suffruticosa Ps-CHI1* gene into tobacco (*Nicotiana tabacum* L.), up to three-fold higher total flavonols and flavones were produced compared to the wild-type control [42]. Furthermore, silencing *CHI* in plants exhibits biological effects that alter total flavonoid and anthocyanin content. For example, transient RNA silencing of *chi* in *Petunia hybrida* resulted in a significant reduction of anthocyanin accumulation in floral tissues [43]. Likewise, VIGS (virus-induced gene silence) targeting *MmCHI1* and *MmCHI2* led to a substantial (about 50%) decrease of approximately 50% in anthocyanin content [44]. In the present study, an *AsiCHI*-overexpression vector was constructed and transferred into *Arabidopsis* ecotype Columbia (Col-0) successfully. The qRT-PCR investigation discovered that the *AsiCHI* gene had significant relative expression in the overexpression lines, while it was not expressed in wild-type plants. The anthocyanin detection results revealed that the transgenic lines had a significantly higher content of anthocyanins than the wild-type. The anthocyanin production was significantly enhanced in transgenic lines. These results indicate that *AsiCHI* activity is essential for in vivo anthocyanin production.

## 5. Conclusions

In this study, the *AsiCHI*, an anthocyanin synthesis-related enzyme gene, has been isolated and characterized from *A. sinicus*. The sequence alignment and phylogenetic analysis of the AsiCHI protein confirmed that AsiCHI is closely related to the other species of the genus *Astragalus* and forms the same group. Moreover, this investigation indicates that the expression of *AsiCHI* was tissue-specific, with high expression in leaves. The functional analysis also revealed that the overexpression of *AsiCHI* played a key role in enhancing anthocyanidin production. These discoveries significantly enhance a deeper understanding of the anthocyanin biosynthesis mechanism in *A. sinicus*.

## Figures and Tables

**Figure 1 genes-14-01400-f001:**
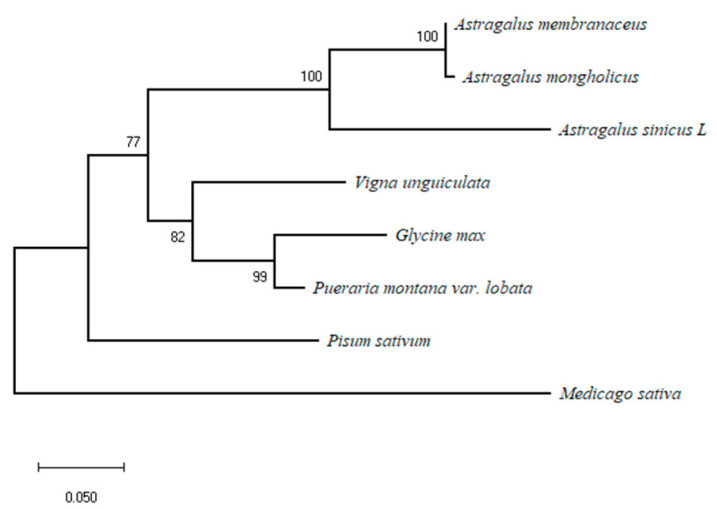
The phylogenetic analyses of AsiCHI and CHIs from different plant species. The Genbank accession numbers are as follows: *Astragalus sinicus* (OQ870547), *Astragalus membranaceus* (ATY39974.1), *Astragalus mongholicus* (ABA55017.2), *Glycine max* (Q53B70.1), *Pueraria montana var. lobata* (ADV71377.1), *Pisum sativum* (XP_050892596.1), *Vigna unguiculata* (QCE05742.1), and *Medicago sativa* (AAB41524.1).

**Figure 2 genes-14-01400-f002:**
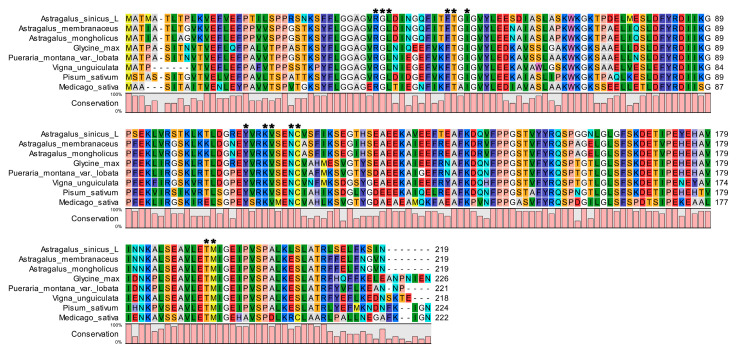
Multiple sequence alignments of the amino acid sequence of CHI. Asterisk represents the active site of AsiCHI. The height of the pink bar under the residues represents the conservation of the respective residue. Other same colors represent the same base. The Genbank accession numbers are as follows: *Astragalus sinicus* (OQ870547), *Astragalus membranaceus* (ATY39974.1), *Astragalus mongholicus* (ABA55017.2), *Glycine max* (Q53B70.1), *Pueraria montana* var. *lobata* (ADV71377.1), *Pisum sativum* (XP_050892596.1), *Vigna unguiculata* (QCE05742.1), and *Medicago sativa* (AAB41524.1). *, *p* < 0.05; **, *p* < 0.01; ***, *p* < 0.001.

**Figure 3 genes-14-01400-f003:**
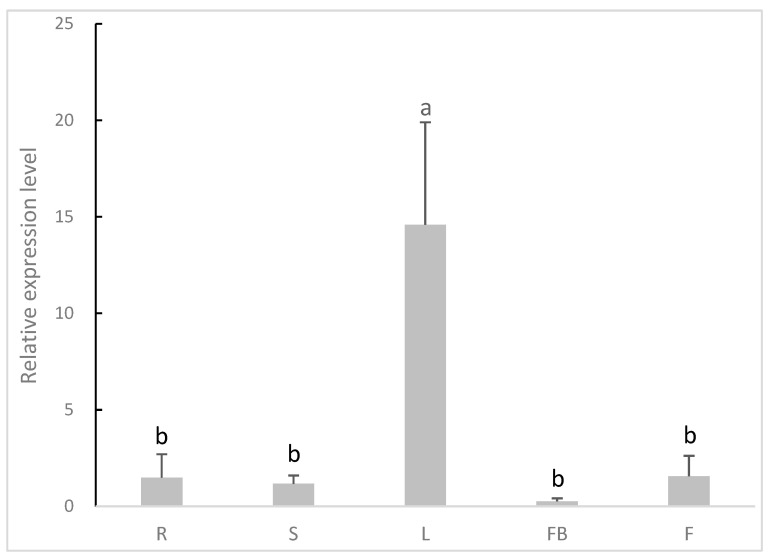
Relative Expression of *AsiCHI* gene in different tissues of *A. sinicus*. Data represent mean ± SD of three biological replicates. Means denoted by the same letter did not differ significantly at *p* < 0.05 according to Duncan’s multiple-range test. Vertical bars indicate the standard deviation of the mean. Roots ®, stems (S), leaves (L), flower buds (FB), and flowers (F).

**Figure 4 genes-14-01400-f004:**
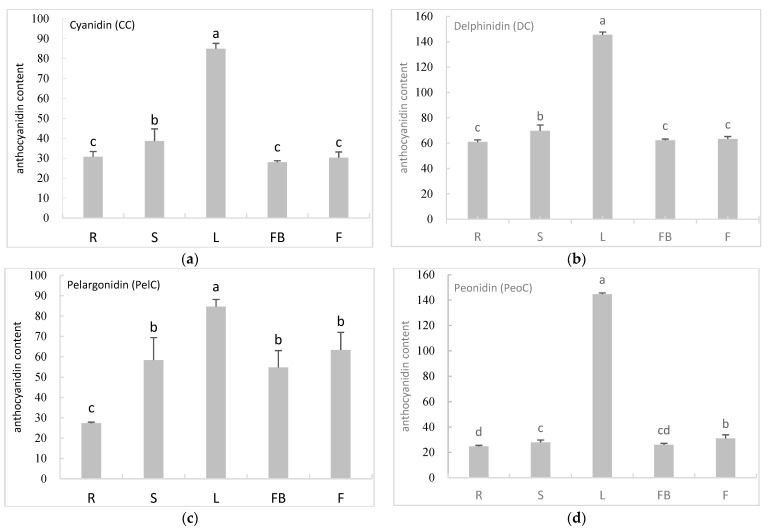
The concentration of six anthocyanidins in different tissues of *A. sinicus*. Data represent mean ± SD of three biological replicates. Means denoted by the same letter did not differ significantly at *p* < 0.05 according to Duncan’s multiple-range test. Vertical bars indicate the standard deviation of the mean. Roots (R), stems (S), leaves (L), flower buds (FB), and flowers (F). (**a**) The concentration of Cyanidin (CC). (**b**) The concentration of Delphinidin (DC). (**c**) The concentration of Pelargonidin (PelC). (**d**) The concentration of Peonidin (PeoC). (**e**) The concentration of Petunidin (PT). (**f**) The concentration of Malvidin (MV).

**Figure 5 genes-14-01400-f005:**
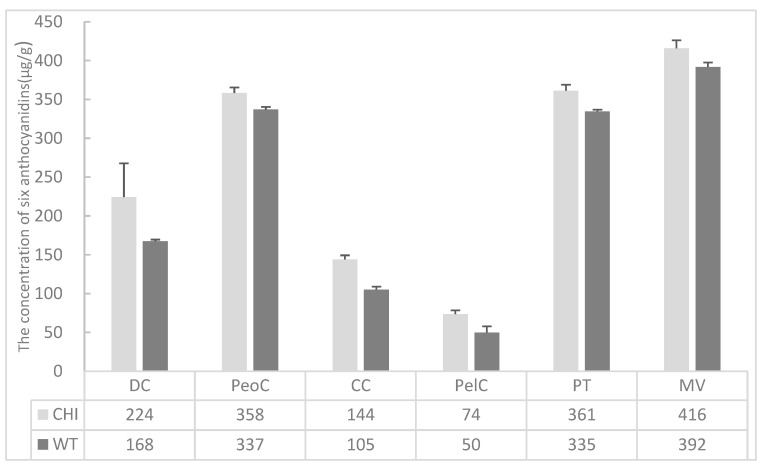
The concentration of six anthocyanidins in *AsiCHI*-overexpression transgenic *Arabidopsis* lines (*CHI*) and wild-type (WT). Data represent mean ± SD of three biological replicates. Vertical bars indicate the standard deviation of the mean. Cyanidin (CC), Delphinidin (DC), Pelargonidin (PelC), peonidin (PeoC), petunidin (PT), and malvidin (MV).

**Table 1 genes-14-01400-t001:** Primer information.

Primer Name	Sequence (5′~3′)	Used to
rCHI-R1	GTCTCCAACACAACCTCAGAAAGTGCCTTGT	5′-RACE amplification
rCHI-R2	CCTGGGGGAAATACTTGATCCTTGAATGCTT	
rCHI-F1	GGGAACACACAGTGAAGCTGAAGA	3′-RACE amplification
rCHI-F2	CCCCCAGGATCTACTGTTTACTACAGACAA	
qCHI-F	AGTCCTTTTTCCTCGGTG	qRT-PCR primers
qCHI-R	GTGATGCTATGTCGCTTT	
Tubulin F	CACCGAGGGAGCAGAGTTGAT	Reference gene for qRT-PCR analysis
Tubulin R	CTGAAACCCTTGAAGGCAGTCA	
HPT-F	GGTCGCGGAGGCTATGGATGC	Detection of transgenic lines
HPT-R	GCTTCTGCGGGCGATTTGTGT	
CHI-F	GGGAACACACAGTGAAGCTGAA	Detection of transgenic lines
CHI-R	GCTCAGATAAGCGAGTAGCCAAA	

**Table 2 genes-14-01400-t002:** Comparison of *AsiCHI* and the orthologous proteins from other species.

Species	NCBI Reference Sequences	No. of Residues	Identity	E Value	PI	Molecular Weight (kDa)
*Astragalus sinicus* L.	OQ870547	219	100	0	5.11	24.14
*Astragalus membranaceus*	ATY39974.1	219	82.65	0	5.4	24.02
*Astragalus mongholicus*	ABA55017.2	219	82.19	0	5.41	23.99
*Glycine max*	Q53B70.1	226	71.1	0	5.27	24.98
*Pueraria montana* var. *lobata*	ADV71377.1	221	71.69	0	5.59	24.13
*Pisum sativum*	XP_050892596.1	224	69.86	0	5.69	24.73
*Vigna unguiculata*	QCE05742.1	218	72.64	0	4.91	24.36
*Medicago sativa*	AAB41524.1	222	59.72	0	5.25	23.83

**Table 3 genes-14-01400-t003:** Correlation of anthocyanins content and *AsiCHI* gene relative expression level.

	Cyanidin (CC)	Delphinidin (DC)	Pelargonidin (PelC)	Peonidin (PeoC)	Petunidin (PT)	Malvidin (MV)
** *AsiCHI* **	0.943 **	0.963 **	0.671 **	0.973 **	0.958 **	0.974 **

**: significantly at *p* < 0.01.

## Data Availability

Not applicable.

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
