# Peer review of "Cloning, Identification, and Functional Analysis of the Chalcone Isomerase Gene from Astragalus sinicus"

_genes, 2023, doi:10.3390/genes14071400_

Round 1

Reviewer 1 Report

This is a valuable article. 

Author Response

Dear Reviewer

We sincerely appreciate your review of our manuscript and your positive evaluation without specific suggestions for improvement. Your support and recognition of our work are truly valued, and they provide us with renewed motivation and confidence in the quality of our research. Thank you for your time and expertise.

Best regards,

Xian Zhang

Reviewer 2 Report

The manuscript reports isolation and organ-specific expression of Astragalus sinensis chalcone synthase. Further, the correlation between biosynthesis of selected anthocyanin and AsCHI expression levels was assessed.

This an interesting study, however there are several aspects that require improvements or explanation:

11)     Please pay attention to the text formatting; in numerous places there is no space among words, various font size is used etc.

22)     The Latin names should be written using italic font;

33)     The full Latin name should be used only at first use, later a shot form should be applied: A. sinensis e.g.;

44)     English language has to be improved;

55)     Materials and Methods:

2.1. it is not clear what was used as a plant material for investigations;

2.4. Which one tubulin gene was used as a reference?

Line 139: method not technique

There is no information on statistical evaluation of data.

3.3. – improve the title of this section;

The correlation between gee expression levels and investigated anthocyanidins should be calculated.

Fig5. – should be Figure 5; further what units are used to present anthocyanidins concentration?

66)     Results:

Table 2: “L” is unnecessary; “var.” should not be written using Italic font;

77)     Conclusions:

“the nucleotide sequence of the anthocyanin synthesis-related enzyme gene the AsiCHI has been isolated and characterized from Astragalus sinicus.” – unclear;

The quality of English language used for manuscript preparation has to be improved, there are some statements that are unclear; spelling also should be corrected.

Author Response

Dear Reviewer,

We greatly appreciate the valuable comments provided by the referees regarding our manuscript titled "Cloning, identification, and functional analysis of chalcone isomerase gene from Astragalus sinicus". The suggestions have proven to be quite helpful, and we have diligently incorporated them into the revised paper. Over the past week, we have extensively consulted relevant literature and papers in order to enhance the overall quality of our work.

On behalf of my co-authors, I would like to address some of the points raised by the reviewer. We have highlighted the revised sentences in red font within the revised paper, aiming to clearly indicate the changes made. Our intention is to ensure that both the reviewers and the editors are satisfied with our responses to the comments and the revisions made to the original manuscript.

Thank you for your time and consideration.

Best regards,

Xian Zhang

11)     Please pay attention to the text formatting; in numerous places there is no space among words, various font size is used etc.

Response: We sincerely apologize for the errors in the paper. We have rectified the text formatting and adjusted the font size accordingly.

22)     The Latin names should be written using italic font;

Response: Thank the reviewer. As per the recommendation, we have now italicized the Latin names in the manuscript.

33)     The full Latin name should be used only at first use, later a shot form should be applied: A. sinensis e.g.;

Response: Thank you for your kind comment. We have adjusted the Latin names as suggested by the reviewer.

44)     English language has to be improved;

Response: We would like to express our gratitude for the reviewer's suggestion. We have diligently enhanced the language quality of the manuscript in accordance with their feedback.

55)     Materials and Methods:

2.1. it is not clear what was used as a plant material for investigations;

Response: Thank the reviewer for the comment. This study used the white petal A. sinicus population selected by our lab as plant materials.

2.4. Which one tubulin gene was used as a reference?

Response: Thank you for bringing this to our attention. To serve as a reference, we utilized the tubulin gene of A. sinicus (OR198065) and have included its corresponding GeneBank accession number in the paper.

Line 139: method not technique

Response: Thank the reviewer’s comment. We have replaced “technique” with “method”.

There is no information on statistical evaluation of data.

Response: We sincerely appreciate the reviewer for the kind comment. All data presented in the paper were repeated in three biological replicates and subjected to analysis using SPSS 26.0. Furthermore, we have incorporated a dedicated "Statistical analysis" section into the revised version of the paper.

3.3. – improve the title of this section;

Response: Thank the reviewer for the suggestion. We have revised the title of the section to “Gene expression profiles of AsiCHI by qRT-PCR”.

The correlation between gee expression levels and investigated anthocyanidins should be calculated.

Response: We agree with the reviewer, and have calculated the correlation of anthocyanins content and AsiCHI gene relative expression level in Table 3.

Fig5. – should be Figure 5; further what units are used to present anthocyanidins concentration?

Response: We apologize for any oversight. The concentration unit used in our study was μg/g, and we have ensured that this unit is properly displayed in the revised paper.

66)     Results:

Table 2: “L” is unnecessary; “var.” should not be written using Italic font;

Response: Thank the reviewer for the suggestion. The font was revised according to the suggestion of the reviewer.

77)     Conclusions:

“the nucleotide sequence of the anthocyanin synthesis-related enzyme gene the AsiCHI has been isolated and characterized from Astragalus sinicus.” – unclear;

Response: We extend our gratitude to the reviewer for their kind comment. We have rewritten this sentence “In this study, the AsiCHI which was an anthocyanin synthesis-related enzyme gene has been isolated and characterized from A. sinicus.”

Reviewer 3 Report

Dear authors

This manuscript is regarding “Cloning and functional analysis of chalcone isomerase gene from Astragalus sinicus”. Authors have characterized the Chalcone isomerase and depicted its potential role in anthocyanins biosynthesis. Additionally, they have shown their role by using a homozygous transgenic line. This manuscript is well-performed and might be a good contribution to researchers working on this research area. Hence, recommended for acceptance after minor revision.

Line 196: However, I have a question regarding the tissues-specific expression, why author chose the relative expression, and which tissue is supposed to be control? I think the author should perform absolute expression for tissue-specific qPCR.

Figure 2: In this figure, the resolution of the image needs to increase, since the image looks blurred.

Kindly check the reference format style in the text according to the author's guidelines.

Author Response

Dear Reviewer,

We greatly appreciate the valuable comments provided by the referees regarding our manuscript titled "Cloning, identification, and functional analysis of chalcone isomerase gene from Astragalus sinicus". The suggestions have proven to be quite helpful, and we have diligently incorporated them into the revised paper. Over the past week, we have extensively consulted relevant literature and papers in order to enhance the overall quality of our work.

On behalf of my co-authors, I would like to address the points raised by the reviewer. We have highlighted the revised sentences in red font within the revised paper, aiming to clearly indicate the changes made. Our intention is to ensure that both the reviewers and the editors are satisfied with our responses to the comments and the revisions made to the original manuscript.

Thank you for your time and consideration.

Best regards,

Xian Zhang

Line 196: However, I have a question regarding the tissues-specific expression, why author chose the relative expression, and which tissue is supposed to be control? I think the author should perform absolute expression for tissue-specific qPCR.

Response: We appreciate the reviewer's comment. The main purpose of the current study is to detect changes in expression levels of CHI in different tissues from Astragalus sinicus, and it is not necessary to accurately determine the copy number of the gene. Therefore, relative quantification was used instead of absolute quantification for detection. Additionally, we have replaced the figure accordingly. It should be noted that the expression level of the Root was intended to serve as the control.

Figure 2: In this figure, the resolution of the image needs to increase, since the image looks blurred.

Response: Thank the reviewer for the kind comment. In order to enhance the resolution, we have submitted the original image of the figure to the editor.

Round 2

Reviewer 2 Report

Dear Authors,

The manuscript has been improved substantially, however it is not still clear if reference gene was gene of α- or β-tubulin?